# Analysis of Nuclear Dynamics in Nematode-Trapping Fungi Based on Fluorescent Protein Labeling

**DOI:** 10.3390/jof9121183

**Published:** 2023-12-11

**Authors:** Liang Zhou, Zhiwei He, Keqin Zhang, Xin Wang

**Affiliations:** 1State Key Laboratory for Conservation and Utilization of Bio-Resources in Yunnan, Yunnan University, Kunming 650500, China; liangzhou@mail.ynu.edu.cn (L.Z.); hezhiwei@itc.ynu.edu.cn (Z.H.); 2Key Laboratory for Microbial Resources of the Ministry of Education, Yunnan University, Kunming 650500, China

**Keywords:** *Arthrobotrys oligospora*, histone H_2_B, EGFP, biological control

## Abstract

Nematophagous fungi constitute a category of fungi that exhibit parasitic behavior by capturing, colonizing, and poisoning nematodes, which are critical factors in controlling nematode populations in nature, and provide important research materials for biological control. *Arthrobotrys oligospora* serves as a model strain among nematophagous fungi, which begins its life as conidia, and then its hyphae produce traps to capture nematodes, completing its lifestyle switch from saprophytic to parasitic. There have been many descriptions of the morphological characteristics of *A. oligospora* lifestyle changes, but there have been no reports on the nuclear dynamics in this species. In this work, we constructed *A. oligospora* strains labeled with histone H_2_B–EGFP and observed the nuclear dynamics from conidia germination and hyphal extension to trap formation. We conducted real-time imaging observations on live cells of germinating and extending hyphae and found that the nucleus was located near the tip. It is interesting that the migration rate of this type of cell nucleus is very fast, and we speculate that this may be related to the morphological changes involved in the transformation to a predatory lifestyle. We suggest that alterations in nuclear shape and fixation imply the immediate disruption of the interaction with cytoskeletal mechanisms during nuclear migration. In conclusion, these findings suggest that the signal initiating nuclear migration into fungal traps is generated at the onset of nucleus entry into a trap cell. Our work provides a reference for analysis of the dynamics of nucleus distribution and a means to visualize protein localization and interactions in *A. oligospora*.

## 1. Introduction

As a large and typical taxonomic group of microorganisms, fungi have had a significant impact on all aspects of human society [1,2,3,4]. Surprisingly, such eukaryotes could survive in space, under extreme temperatures, at extreme altitudes, and even in Chernobyl and Fukushima radioactive reactors, demonstrating their strong environmental adaptability [5,6,7]. Based on their astonishing growth characteristics, people have applied them to various biotechnological and industrial applications and biological control fields [8,9,10]. Plant pathogenic nematodes could cause global crop losses of billions of dollars annually, and are one of the important reasons for food security issues [11,12]. Nematode-trapping fungi (NTF) are an effective means of controlling plant pathogenic nematodes [13,14]. However, our understanding of the entire life cycle involved in the transformation of numerous fungal forms is quite limited. Dynamic changes in the nucleus could significantly affect the life cycle of fungi [15]. Throughout the entire life cycle of fungi, many activities of hyphae depend on the precise regulation of the nucleus [16,17]. The nuclear dynamics of fungi depend on complex genetic regulatory systems and cytological events [18]. Understanding these nuclear dynamics provides a means to understand the growth law of fungi and strengthen the application of biotechnology.

Fungi occupy an important niche due to their species diversity and growth tolerance [19]. Predation is a common phenomenon in nature [20], and NTF and nematodes are good examples due to their low research costs and easy observation of results [21]. When sensing the presence of nematodes, NTF could quickly generate trapping devices to lure and capture nematodes [22]. Nematode-trapping fungi exhibit diverse morphologies in their predatory organs, and molecular phylogenetic analysis indicates that the majority of NTF belong to the phylum Ascomycota. Moreover, research reveals that the predatory organs of NTF primarily evolve in two directions: one involves the formation of constricting rings, while the other leads to the development of three-dimensional fungal nets, adhesive knobs, and adhesive columns, constituting adhesive trapping structures [14]. Considering the harm of pathogenic nematodes and the potential application of NTF as biocontrol agents, it is particularly important to fully understand the dynamics of nematode-trapping fungi. *Arthrobotrys oligospora* is widely distributed around the world [23]; it is a typical type of NTF and is often used in the field of biocontrol [24], which has been accompanied by the development of relevant genetic manipulation methods [25]. Similar to other NTF, the dynamic development process of mycelium is crucial for capturing prey [26]. The key stages of this dynamic development process include spore germination, hyphal extension, production of traps, and, ultimately, digestion of prey. Surprisingly, little is known about the nuclear dynamics of the aforementioned biological processes, which play a crucial role in the predation of nematodes by *A. oligospora*.

Artificial cultivation of *A. oligospora* is easy to carry out, and a significant feature is that it could undergo morphological transformation in a short period of time in a laboratory environment [27]. The lifestyle of *A. oligospora* begins with the release of mature conidia from its conidiophore, and it is known that spore germination is induced by certain stimulants, including amino acids, water, and temperature [24]. When external conditions are suitable and spore dormancy is broken, a series of significant morphological changes occur, leading to the germination of conidia and the formation of nutrient hyphae. Interestingly, when *A. oligospora* perceives nematode signals such as ascarosides, its mycelium undergoes drastic and rapid morphological changes, becoming specialized as a trap [28]. The initial growth of trap cells is manifested as branching hyphae that are perpendicular to the vegetative hyphae. During the process of elongation and curling, these branching hyphae gradually form a structure of multi-cells, which eventually fuses with the parental vegetative hyphae to form three- or four-cell loops capable of capturing nematodes. It is worth noting that the relevant adhesion substances play an important role in assisting its traps in catching nematodes [29,30,31,32,33]. The morphological transformation from hyphae to traps represents a transformation in this NTF’s lifestyle. Recently, in research on the perception of nematode signals by *Arthrobotrys flagrans*, nuclear fluorescence intensity has been used as an indicator [34]. In addition, fluorescent proteins are used to trace the growth of *Arthrobotrys flagrans* hyphae [35,36,37]. However, research on the nuclear dynamics of nematode-trapping fungi is still lacking.

The co-existence of nuclei with different genetic backgrounds and the regulatory network of nucleus movement between septa are typical characteristics of multi-nucleated filamentous fungi [38]. These characteristics make nematode-trapping fungi good research objects for studying the dynamics. Fluorescent labeling is a good technical tool for studying the behavior and dynamics of living cell nuclei, and related model approaches have also been established [39]. The emergence of germ tubes and the formation of septa are precisely controlled, and this has been extensively studied in *Aspergillus nidulans*, which has been proven to be closely coordinated with nuclear division [40,41,42,43,44]. On the contrary, studies have shown that the production of germ tubes in the macroconidia of *Fusarium graminearum* is not related to nuclear division [45,46]. In *Sordaria fmicola*, studies have shown that during pre-contact interactions, the nucleus of newly germinated hyphae and fused hyphae is located near the tip [47]. In terms of nuclear dynamics research on *Neurospora crassa*, it has been reported that it is related to the dynamic changes in microtubules [48,49]. Furthermore, during the spore germination and hyphal fusion process of *Fusarium oxysporum*, nuclear dynamic changes are closely related to the establishment of infection [50,51].

Although *A. oligospora* has long been used to study various aspects of how fungi capture nematodes, nuclear fluorescence-labeled strains produced through genetic transformation methods have not been established. Therefore, establishing a practical and feasible live cell tracing method with specific fluorescence localization may address the shortcomings of *A. oligospora* spontaneous fluorescence. To explore the nuclear dynamics of different developmental stages of *A. oligospora*, we generated three histone H_2_B–EGFP-labeled strains, allowing us to conduct live cell analysis of the nuclear dynamics of this species for the first time. In addition to providing visual tools for gene function research, the time-lapse video provides an opportunity to objectively observe and gain a deeper understanding of the specific behaviors exhibited by the nucleus of *A. oligospora*.

## 2. Materials and Methods

### 2.1. Strains, Media, and Culture Conditions

*Arthrobotrys oligospora* (ATCC 24927) was obtained from the Key Laboratory for Conservation and Utilization of Bio-resource, Yunnan University, and used as the experimental research strain in this study. *A. oligospora* was grown on a common fungal nutrient medium (potato dextrose agar, PDA). TG liquid medium (1% tryptone, 1% glucose) was used for obtaining young mycelia; TB3 medium (20% sucrose, 0.3% tryptone, 0.3% yeast extract, 0.75% agar) was used for the growth of transformants; TYGA medium (1% tryptone, 0.5% yeast extract, 1% glucose, 1% molasses, 2% agar) was used for the rejuvenation of transformants; and CMY medium (2% corn flour, 0.5% yeast extract, 1.5% agar) was used for enriching conidia, while conidia were germinated on water agar medium (1.5% agar). All fungal culture media mentioned above were cultured at 28 °C. The nematode *Caenorhabditis elegans* N2 was maintained on nematode growth media (NGM) at 20 °C. *Escherichia coli* DH5α strains were maintained on Luria–Bertani (LB) medium, and *E. coli* harboring the targeted pJET1.2 plasmids were selectively cultured on LB medium containing 100 µg/mL ampicillin at 37 °C.

### 2.2. Genome Extraction and Analysis

The extraction protocol for *A. oligospora* genome was slightly improved on a previous method [52]. Firstly, the strains were grown on PDA medium for 3–4 d, and the mycelia were ground into powder using liquid nitrogen. Subsequently, the DNA extraction reagent [phenol/chloroform/isoamyl alcohol (*v*/*v*/*v*) = 25:24:1] was added to the mycelium powder after sufficient vortex and then incubated for 1 h at 65 °C. The supernatant was obtained via centrifugation at 12,000 rpm for 10 min, and an equal volume of isopropanol and one-tenth volume of sodium acetate (3 M) were added to the supernatant, which was precipitated at −20 °C for 30 min. The pellet was obtained via centrifugation at 12,000 rpm for 10 min again, washed twice with 70% ethanol, and finally collected in a centrifuge tube. The DNA pellet was placed in air to dry and finally resuspended using sterile ddH_2_O.

### 2.3. Construction of Fluorescent Plasmids

To construct a nuclear fluorescent plasmid, the pJET1.2/blunt-EGFP plasmid was amplified by means of fusion polymerase chain reaction (PCR) using Phanta Max Super-Fidelity DNA Polymerase (Vazyme, Nanjing, China) according to the manufacturer’s protocol. The *A. oligospora* H_2_B promoter/gene (H_2_Bp/H_2_B) and H_2_B terminator (H_2_Bt) fragments were amplified with the primer pairs F1/R1 and F4/R4, respectively. EGFP and hygromycin-resistance cassette (HygR) fragments were then obtained via amplification with F2/R2 from the pCT74 vector and F3/R3 from the pFC332 vector, respectively. The amplified fragments were cleaned and purified using the HiPure Gel Pure DNA Mini Kit (Magen, Guangzhou, China). The fusion fragments of H_2_Bp/H_2_B–EGFP and HygR–H_2_Bt were obtained via amplification with F1/R2 and F3/R4, respectively. These two fragments were purified, mixed, and amplified with the primers F1 and R4 to generate H_2_Bp/H_2_B–EGFP–HygR–H_2_Bt fragment. The overlap-extension PCR product was cloned into the pJET1.2/blunt vector (Thermo Scientific™, Waltham, MA, USA) and, finally, pJET1.2/blunt-H_2_Bp/H_2_B–EGFP–HygR–H_2_Bt was obtained. For PCR screening, the reaction was carried out using 2 × Taq Plus Master Mix II (Vazyme, Nanjing, China) with YZ-F/YZ-R. The primer sequences and strains are listed in Appendix A.

### 2.4. Arthrobotrys Oligospora Transformation and Verification

*A. oligospora* transformation was carried out using a polyethylene glycol/CaCl_2_-mediated protoplast approach with slight modification, as described previously [52]. In brief, *A. oligospora* grew for 3–4 days in a 6 cm culture plate with PDA medium. When mycelia were not fully grown on the entire plate, we transferred the mycelial plugs of PDA to TG liquid medium at 28 °C for 2 days without shaking. Subsequently, the TG liquid medium was placed at 28 °C and shaken at 180 rpm for 10 h. Young mycelia were harvested via filtration and subsequently washed with MN buffer (0.3 M MgSO_4_, 0.3 M NaCl). Protoplasts were generated by digesting with a lysing buffer composed of 10 mg/mL of cellulase (Sangon Biotech, Shanghai, China) and 8 mg/mL of snailase (Solarbio, Beijing, China) in the MN buffer at 28 °C for 3–4 h. In order to obtain protoplast cells, after removing the supernatant via centrifugation with the lysing buffer, the supernatant was washed with KTC buffer [1.2 M KCl, 10 mM Tris-HCl (pH = 7.5), 50 mM CaCl_2_] and centrifuged at 5000 rpm (4 °C for 5 min) to collect protoplasts, and the final concentration was adjusted to 10^6^–10^7^ cells/mL KTC buffer [53]. The corresponding plasmid was added to the KTC buffer-resuspended protoplasts, mixed well, and then ice soaked for half an hour. Subsequently, 1 mL of PTC buffer [KTC buffer: 40% PEG4000 (*v*/*v*) = 1:2] was added to the EP tube and incubated at 28 °C for half an hour. The mixtures were plated on TB3 medium and incubated at 28 °C for 16 h. Then, each plate was covered with 5 mL of TB3 medium supplemented with 100 µg/mL hygromycin B and incubated at 28 °C for 5–7 days. The colonies of the transformants were transferred to TYGA medium containing 100 µg/mL hygromycin B. The positive transformants were confirmed via PCR, Southern blotting, and fluorescence signaling, as described previously [52]. The relevant primers are listed in Appendix A.

### 2.5. RT-qPCR Analysis

WT and positive transformants were cultured on PDA at 28 °C for 5 days, the mycelium block was inoculated into TG liquid culture medium, and the mycelia were collected after 3 days of cultivation at 28 °C. The mycelial samples were collected for total RNA extraction according to the Sangon kit’s procedure (Sangon Biotech Company, Shanghai, China) and then reverse transcribed into cDNA using a PrimeScript RT reagent kit (Vazyme, Nanjing, China). They were then used as the template for qPCR. Specific primers (Appendix A) were used to detect the transcript levels of the *EGFP* gene, and the *A. oligospora β-tubulin* gene (*AOL_s00076g640*) was used as an internal standard using a Roche LightCycler 480 relative quantitative method (Roche Center, Shanghai, China). The relative transcription level (RTL) of each gene was calculated as the ratio of the transcription level between the positive transformants and the WT strain according to the 2^−ΔΔCT^ method [54].

### 2.6. Microscopic Observations

The conidia of *A. oligospora* wild-type and fluorescence-labeled strains were incubated on a WA plate (6 cm Petri dish) and cultured at 28 °C. Photographs of the conidia, hyphae, and traps were taken using glass slides. The dynamic observation of the culture at different time points was carried out using dual channels of fluorescence and white light. For fluorescence microscopy, the images and time-lapse videos of live cells were captured with the ECLIPSE LV-100ND Optical System equipped with a CMOS camera on a Nikon DS-Ri2 inverted microscope, using a tube lens with a focal length of 200 mm, an objective with a parfocal distance of 60 mm, and an objective thread size of 25 mm. The EGFP images were acquired using standard filter sets of 475/25 and 510/20 nm for excitation and emission, respectively [55]. The images were analyzed using NIS-Elements Viewer 5.21 (Applied Precision, Shanghai, China).

### 2.7. Verification of Fluorescence Localization

To observe fluorescence localization, calcofluor white (Sigma-Aldrich, St. Louis, MO, USA) and 4′,6′-diamidino-2-phenylindole (DAPI; Sigma-Aldrich) were used to stain the conidia, mycelia, and traps of positive transformants, as described previously [56]. To prove that EGFP had been inserted into the genome of *A. oligospora*, the positive transformants were confirmed via PCR and Southern blotting, as described previously. Furthermore, the total, nuclear, and cytoplasmic proteins of *A. oligospora* were extracted separately, following the protocol of the protein extraction kit used (Bestbio, Shanghai, China). Proteins were resolved with 12% sodium dodecyl sulfate–polyacrylamide gel electrophoresis and underwent immunoblot analysis with GFP rabbit monoclonal antibody (Beyotime, Shanghai, China) and H_2_B rabbit monoclonal antibody (Majorbio, Shanghai, China). Protein bands were visualized using an EasySee^®^ Western Blot Kit (TRANS, Beijing, China) in a chemiluminescence imaging system (Amersham ImageQuant™ 500 CCD, Shanghai, China).

### 2.8. Comparison of Phenotypes between WT and Positive Transformants

To compare the growth of mycelia under different nutritional conditions, we cultivated them on five culture media: barren nutrient culture medium CMA; basic culture medium PDA; and eutrophic culture media TYGA, TG, and CMY. We recorded the growth rate of mycelium at 24 h intervals. To count the number of conidia, the fungal strains were inoculated on CMY medium at 28 °C for 10 days, and the spores were washed from the medium, as described previously [52]. A total of 300 spores were spread on a 9 cm plate with water agar medium. The spore germination rate was counted at 4, 8, and 12 h. Similarly, 3 × 10^4^ spores were spread on a 9 cm plate with water agar medium at 28 °C for 3 days. After spore germination, approximately 200 *C. elegans* were added to each plate for the induction of traps formation. After 24 h of adding nematodes, we counted the number of traps and estimated nematode mortality rates.

### 2.9. Statistical Analysis

Graphs in this study were produced utilizing GraphPad Prism 9.0.0 software, and statistical significance was assessed through a *t*-test employing the same software version, namely, GraphPad Prism 9.0.0. Image J software was used to analyze images from both Western blot and Southern blot. For fluorescence localization analysis, laser confocal microscopy was used (Olympus, Tokyo, Japan, FV3000).

## 3. Results

### 3.1. Generation of Nuclear-Labeled Strains

To study the nuclear dynamics of *A. oligospora*, our aim in this study was twofold. Firstly, we aimed to distinguish spontaneous fluorescence through the special nuclear localization of H_2_B, and, secondly, we recorded real-time images of nuclear distribution and behavior at different stages of the life cycle. As important structural proteins for chromatin assembly, histones play crucial roles in maintaining high conservation across all living cell types [57]. We used the fluorescent EGFP protein to tag H_2_B histone (*AOL_s00169g5*) to monitor nuclear distribution during the conidial germination, hyphal extension, and trap formation processes. As shown in Figure 1A, we successfully constructed the histone H_2_B-labeled strains using PEG/CaCl_2_-mediated protoplast transformation. Using the hygromycin B resistance gene as a screening marker, the positive transformants exhibited strong GFP signals under fluorescence microscopy (Figure 1B). Four transformants were further screened using PCR to detect the presence of H_2_Bp/H_2_B–EGFP–HygR–H_2_Bt fragments. Specifically, using the four transformants’ genomes as templates, primers were used to amplify fragments of 2395 bp, 2607 bp, and 3444 bp from F1/R2, F2/R3, and YZ-F/YZ-R of these transformants, respectively. However, the corresponding bands could not be amplified in the wild type. Using a universal F4/R4 primer pair as a control to amplify gDNA, a product of 1002 bp was produced in both the wild-type and transformed strains (Figure 1C). The Southern blot analysis (Figure 1D) revealed that three transformants integrated a single copy, and the presence of the same band sizes indicated that the H_2_Bp/H_2_B–EGFP–HygR–H_2_Bt fragment was inserted into the genome. Among them, the probe was designed using EGFP sequence. Using YZ-F/YZ-R amplification as before, all strains, except for the wild-type strain, could amplify the 3444 bp bands, indicating that EGFP was indeed inserted into the genome (Figure 1C). Therefore, we used transformants one, two, and four for subsequent experiments. The RT-PCR results show a significant increase in the expression of EGFP in these three transformants (Figure 1E). The Western blot results show that EGFP is only specifically expressed in the nucleus of all three strains. Furthermore, we selected a strain and used H_2_B as the internal reference to verify that there was no cross-contamination of the protein (Figure 1F). In summary, we successfully constructed EGFP-labeled H_2_B strains, which provide a basis for detecting the nuclear dynamics of *A. oligospora*.

### 3.2. Comparison of Phenotype Analysis between the Transformants and WT

To present nuclear dynamics as realistically as possible, we analyzed the phenotype of the transformants compared to the WT. First of all, accurate localization of fluorescent proteins is crucial for subsequent research; thus, we analyzed the transformants via CFW, DAPI, and FM4-64 staining [58,59]. The results show that CFW clearly labels the septum and cell wall, and EGFP fluorescence signal has obvious co-localization with DAPI. Interestingly, the cytoplasm labeled with FM4-64 shows partial voids in the location of nucleus (Figure 2A). Nematophagous fungi play an excellent regulatory role in the biological control of crop pathogenic nematodes in nature, and the formation of traps is crucial to such control. In the presence of *Caenorhabditis elegans*, the transformants could produce a large number of traps (Figure 2B), which exhibited strong catching ability (Appendix A). Furthermore, the mycelia penetrated the insect body and digested the nematode (Figure 2C and Appendix A). The scanning electron microscopic (SEM) results also indicate that the mycelia does indeed invade the insect body (Appendix A). By using a laser confocal microscope to scan different layers of the traps, it was found that fluorescence appeared on different planes, further indicating the three-dimensional structure of the traps (Figure 2D and Appendix A). The typical feature of such a trap is a circular structure composed of multiple cells, and the CFW staining results further support this phenomenon (Appendix A). The above results indicate that the transformants are consistent with the WT in terms of trap formation and nematode capture.

On this basis, we continued to measure multiple indicators such as mycelial growth rate, number of conidia, spore germination rate, and production of traps. In terms of mycelial growth rate, there was no significant difference in the growth rate of the transformants and the WT (Appendix A). In addition, our statistical results showed that there were no significant differences between the transformants and the WT in terms of spore number, spore germination rate, number of traps, and nematode killing activity (Appendix A). To further investigate whether the morphology of the mycelium changed, we used *Caenorhabditis elegans* to induce the fungus to produce traps (Appendix A). The SEM results show that there are no significant differences in terms of the morphology of conidia, hyphae, and traps when compared to the WT (Appendix A). In addition, we further observed the formation process of the transformants’ traps and found no difference compared to the WT. Specifically, we observed three situations: immature traps continued to form a three-dimensional fungal loop (Appendix A, left), traps formed on young hyphae after spore germination (Appendix A, middle), and mature traps formed on hyphae (Appendix A, right). Interestingly, we found that after 48 h of spore germination, strong GFP signals still existed in the hyphae and traps, indicating that the transformants we obtained could stably express exogenous fluorescent proteins (Appendix A). Based on the above research results, it is concluded that the transformants could truly reflect the nuclear dynamics of the WT.

### 3.3. Nuclear Distribution during Conidia Germination in A. oligospora

Conidia, as important propagules of *A. oligospora*, grew in clusters on the stem of conidiophore (Appendix A). When the conidia had not yet germinated, a large number of nuclei appeared within them, which distinguished the spore septum, as indicated by CFW staining and white light (Figure 3A). In addition, we found that the spores of *A. oligospora* mostly sprouted a mycelium and occasionally two or three hyphae appeared (Appendix A). To detect the nuclear dynamics in the conidia, the conidia of the *A. oligospora* strains were first collected and inoculated on the basic culture medium. Approximately 1–2 h after inoculation, the conidia of the H_2_B–GFP-labeled strain formed pores at the end of spores, followed by the formation of germ tubes, which allowed the nucleus to enter (Figure 3B). After germination, the germ tubes that emerged from the conidia continued to elongate and branch, producing young vegetative hyphae. It is worth noting that the nucleus appeared in the germ tubes just after the spores germinated (Figure 3C), which is completely different from the nuclear dynamics involved in the germination of both *Neurospora crassa* and *Sordaria fmicola* [47,60]. Further, 5 h after inoculation with conidia, more cell nuclei appeared at the tip of the hyphae (Figure 3D). Interestingly, we found that 24 h after spore germination, a large number of nuclei appeared in the mycelia (Figure 3E). The movement of the nuclei could be observed in the hyphae and conidia, indicating that there may be some connection between them (Appendix A). Similarly, the phenomenon of nucleus appearing at the tip of germ tube has been reported in various filamentous fungi, including *N. crassa*, *F. oxysporum*, and *S. fmicola* [47,50,61,62,63]. In addition to that, nuclei are transported into germination vesicles that form on one side of the spores [64], which is consistent with our research findings.

### 3.4. Nuclear Behavior in Mature Vegetative Hyphae

After the spores germinated to form mature hyphae, we observed the distribution of the nuclei along the length of the nutrient hyphae (Appendix A). During the growth of vegetative hyphae, nuclear dynamics are complex. Unlike yeast [65], the most typical process is the process of nuclear division, and we recorded the dynamic changes in nuclear division in *A. oligospora*. It is worth noting that there was no nucleus around the divided nucleus, and with nuclear division, the new nucleus gradually migrated to a non-nuclear area. In addition, the other nuclei in the mycelium show dynamic changes and, finally, show a more uniform state in the mycelium (Figure 4A, Appendix A). Interestingly, we found that the movement of nuclei in the mycelium was not entirely dependent on neighboring nuclei. Specifically, a certain nucleus could cross adjacent nuclei and migrate toward the tip of the hyphae, ultimately becoming the nucleus closest to the tip of the hyphae (Figure 4B, Appendix A). We believe that this may be related to mycelial growth, as reported in *Neurospora crassa*, whose cytoplasmic bulk flow can promote nuclear movement [48,49,66,67]. Similarly, we observed reverse nuclear migration in mature and branched hyphae. The nucleus first migrated toward the tip of a mature hypha, and then migrated in the opposite direction into a branching hypha. In branched hyphae, there was also a phenomenon of bidirectional migration of the nucleus, which first migrated toward the tip of a branched hypha, then moved in the opposite direction, and finally left the branched hypha (Appendix A). In addition, we also observed a cross-nuclear migration phenomenon, similar to Figure 4B, in branching hyphae, but it did not occur at the tip of the hyphae (Figure 4C, Appendix A). Additionally, we found that there was a phenomenon of nuclear migration toward the tip of rapidly growing hyphae. The speed of nuclear movement was extremely fast, and when a large number of nuclei accumulated at the tip of the mycelium, the mycelium also regulated the movement of excess nuclei away from the tip (Figure 4D, Appendix A). In summary, the dynamic changes in the nucleus of a mycelium exhibit precise regulatory activities.

### 3.5. Nuclear Distribution during Trap Formation in A. oligospora

The research on traps is fascinating, as our previous studies have shown that in addition to nematodes, ammonia can also induce the production of traps [68]. Based on previous research findings, and to further understand the localization of the nucleus in these traps, we first observed the different stages of trap formation. At the beginning, when the mycelium began to bend, the nucleus entered it, but there was no nucleus at the tip of the curved mycelium (Figure 5A). When the trap grew to a half loop, there was a large number of nuclei present in the trap, and these nuclei could be observed at the tip of the hyphae (Figure 5B). As the trap grew further, the nuclei could still be observed at the tip of the mycelium (Figure 5C). Until the end, when the trap had matured, a large number of closed circular fungal loops were formed, and the nuclei were evenly distributed within them (Figure 5D). Consistent with the phenomenon of nuclei at the tip of hyphae, there was also a nucleus at the apex of the loop-shaped cell extension during the formation of the trap. We believe that this type of hyphal or loop-shaped cell tip nucleus plays an important role in hyphal extension, signal recognition, and other aspects.

As the unique predatory organ of *A. oligospora*, how do loop-shaped cells form a spatial three-dimensional structure (trap)? We observed the traps shown in the red dashed box of Figure 5D and found that independent looped traps could be connected through hyphae (Figure 6A). On the other hand, new looped traps could grow on old looped traps, which we observed on the H_2_B-positive transformants via CFW staining (Figure 6B, Appendix A, right). Based on the two connection modes of loop cells mentioned above, a spatial stereoscopic trap could be formed (Figure 6C).

## 4. Discussion

For many decades, the division and migration of fungal nuclei has attracted the interest of researchers [69,70,71,72]. Structural imaging analysis under non-living cell conditions, as examined via scanning electron microscopy (SEM), highlights the complexity of nuclear structure [73]. However, the lack of living cell biology hinders researchers’ understanding of nuclear dynamics during processes such as spore germination and hyphal extension and fusion [63]. In early studies on nematode-trapping fungi, following the work of Woronin, who first observed the predatory organs of NTF, Drechsler discovered in 1934 that these predatory organs included an adhesive net while he was observing non-contractile loops. In particular, Drechsler provided a detailed description of a large number of NTF and drew exquisite images. Many NTF’s names are related to their discoverers’ names [74,75,76]. Therefore, the discovery of nematode-trapping fungi is not only a scientific miracle but also shows that these fungi are a valuable resource that are widely distributed in natural soil, agricultural soil, and various organic residues. Nearly half a century after the publications of these pioneering research works, with advancements in the analysis of the relevant mechanisms for capturing nematodes, numerous NTF products have been applied in the field of biological control [77,78,79].

In this study, we proved that nuclei could indeed migrate rapidly in hyphae, and they can originate from different conidia. In addition, we often observed the migration of cell nuclei in the opposite direction to the tip of hyphae, indicating that the regulation of cell nucleus migration by filamentous fungi is reversible. To this end, we tracked the phenomenon of nuclear migration in mature and branched hyphae, which is pulled by certain components, as evidenced by the presence of comet-like fluorescence-labeled nuclei (Appendix A). Although these observations require further experimental confirmation, we believe that the lifestyle changes occurring in fungi in order to capture nematodes is related to the rapid migration of cell nuclei, as the phenomenon of ‘immovable’ fungi capturing rapidly moving nematodes is common. The mechanism of trap formation may remind people of the organs of plants for capturing prey, such as those of pitcher plants [80]. It is worth noting that the traps of fungi are induced, and the rapid migration of nuclei is evolutionarily understandable as a mechanism for facilitating the rapid growth of mycelia and the rapid formation of traps [81].

Although we do not know how the nucleus of a mature hypha enters a branching hypha by crossing the septum, we observed that the nucleus first hesitates to turn at the T-junctions and then turns to another branch, which may be the correct path to reach the branching hypha. In terms of the phenomenon of the nucleus moving back and forth in the mycelium in a retrograde and anterograde manner, we speculate that the nucleus establishes a certain connection with a relevant molecular motor, that is, the precise direction of the nucleus’s movement is regulated by the molecular motor.

Whether there are other regulatory methods for the entry of a nucleus into a trap during the growth process of the trap remains to be solved [66]. After spore germination, the nuclei mostly gather at the tip of the hyphae. When observing the spores and their germinating hyphae, the nuclei exhibit significant oscillatory motion near the tip of the hyphae. They work in an astonishing synergistic manner, wherein the nuclear movement in the mycelium and the nuclear movement in the spores synchronize (Appendix A). Identifying the factors that control the movement of nuclei during the formation of mycelium elongation will be a huge challenge. The first clue is given by observing the shape of these nuclei in the different steps mentioned above. In fact, we observed that the shape of the nuclei includes circular, pear-shaped, and comet-shaped, and the shape changes again when the nuclei move. These changes may remind us to pay close attention to the interaction between the cell nucleus and cytoskeleton [82]. In filamentous fungi, cytoskeleton and molecular motors, such as kinesins and dynein/dynein complex, precisely regulate the movement of the nucleus [83,84]. Especially when cytoskeleton-depolymerizing drugs are used to treat cells, changes in shape and oscillatory motion are affected [85,86]. In addition, the nucleus in a ropy mutant has a rounded shape, which is completely different from the oval- or pear-shaped nucleus in wild-type cells [87]. Therefore, the oscillatory movement of nuclei in hyphae may involve cytoskeleton and molecular motor mechanisms. We speculate that the changes in the shape of nuclei and their fixation indicate the instantaneous loss of the interaction between these nuclei and cytoskeleton mechanisms during nuclear migration. In summary, these results allow us to assume that the signal that triggers nuclear migration into fungal traps is generated at the beginning of the entry of the nucleus into a trap cell.

By dynamically observing fluorescence imaging [88], our work modestly aims to solve the evolutionary issues concerning the fate of the nucleus during the development of NTF. On the other hand, our study broadens current perspectives by revealing an unexpected behavior of the nucleus during mycelial development, which is not observed in simple research based on DAPI staining. Our live-cell imaging method for studying hyphal growth in NTF has raised new issues: What is the mechanism by which the nucleus is pulled during mycelial elongation? What signals guide the dynamic changes in the nucleus in a trap? What is the mechanism regulating the relatively fixed number of nuclei in a single cell between two septums? Is there any way to regulate the number of cell nuclei crossing the septum? On top of this, the influence of the non-deformation of the nucleus in curved hyphal cells during the formation of traps may provide a new direction for subsequent research. Numerous human diseases involve mechanical signal transduction failures, and studies have shown a positive correlation between nuclear deformation and cell invasion in malignant tumors [89]. Therefore, studying the chromatin differences caused by the deformation of *A. oligospora* trap nuclei may lay the foundation for exploring gene expression patterns in eukaryotes.

Nematophagous fungi are widely used in biological control [90]. Different types of nematophagous fungi employ varying mechanisms when acting on nematodes. However, a common feature among them is the necessity to breach the nematode’s body wall to exert their lethal effects. As a result, toxic factors such as serine proteases, chitinases, small-molecule-secreted proteins, and secondary metabolites produced by these fungi play crucial roles in nematode control [91]. *V. chlamydosporium* germinates from spores to produce hyphae. Upon contact with nematode eggs, it punctures the eggshell, subsequently degrading the eggshell, vitellin, chitinous layers, and protoplasm. This process ultimately leads to the death of female nematodes and their eggs [92]. On the other hand, *P. lilacinus* spores germinate in the soil to generate hyphae. These hyphae, upon encountering nematode eggs, secrete chitinases to break down chitosan in nematodes and the egg’s oligosaccharide shells. This results in the dissolution of the nematode body wall and eggshell. Subsequently, the hyphae proliferate abundantly using the internal nutrition of the egg as a food source, disrupting nematode eggs and inhibiting the hatching of nematode larvae [93]. Building upon this foundation, specific types of fungi that consume nematodes are employed in various domains, including agriculture, animal husbandry, and plant protection.

In a more detailed context, the incorporation of specific fungi into animal feed serves as a targeted strategy to manage and regulate the population of nematodes within the digestive systems of animals. By introducing these fungi into the feed, a natural and sustainable approach is employed to control internal parasitic nematodes in livestock, enhancing the overall health and productivity of the animals. On the plant side, these fungi play a crucial role in soil-borne pest management. When applied to the soil, they act as biocontrol agents, capturing and suppressing plant-pathogenic nematodes. This application not only aids in safeguarding the health of plants but also contributes to sustainable agriculture practices by reducing the reliance on chemical pesticides. The overall objective is to achieve effective plant protection while minimizing the environmental impact associated with conventional pest control methods. With the growing influence of modern environmental protection and ecological civilization, nematode-trapping fungi have garnered widespread attention as a biological control method against parasitic nematodes [94]. Significant progress has been made in various aspects concerning these fungi, such as strain identification through fungal resource surveys, efficient selection of biocontrol fungal strains, formulation of fungal agents, and their practical application in fields [95]. However, despite these advancements, current practices in pest control still face several pressing issues. For instance, there is a lack of highly effective and stable biocontrol fungal strains. Understanding the colonization of nematode-trapping fungi in diverse environments, elucidating the molecular mechanisms underlying their action on nematodes, and comprehending the molecular responses of nematodes to these fungi are areas that require further exploration and research. These gaps in knowledge pose challenges that need to be addressed for more effective and comprehensive implementation of nematode-trapping fungi in pest management strategies.

## 5. Conclusions

In this study, a fluorescence labeling system was established for *A. oligospora*. Our real-time imaging reveals a consistent spatial pattern in living cells, with nuclei of germinal and elongated hyphae positioned near the cellular tip, emphasizing the dynamic interplay between nuclear positioning and fungal cell growth. Additionally, the unexpectedly rapid migration speed of cell nuclei challenges conventional expectations, prompting a reevaluation of factors influencing nuclear mobility in mycelium and offering insights into broader intracellular transport process. Further efforts should be made to utilize fluorescent labeling systems for gene function research.

## Figures and Tables

**Figure 1 jof-09-01183-f001:**
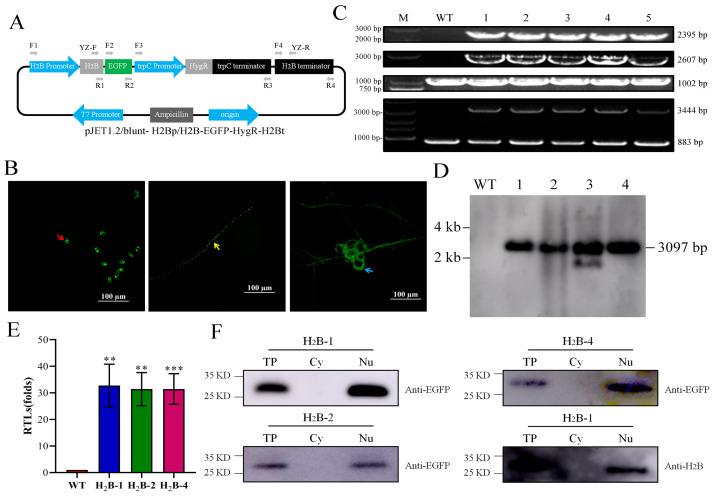
Generation of fluorescently labeled *A. oligospora* strains. (**A**) Schematic presentation of the plasmid construct used to generate histone H_2_B–EGFP-labeled strains. (**B**) Fluorescence observation of conidium, hyphae, and traps. Red arrow: conidium. Yellow arrow: hyphae. Blue arrow: traps. Bar: 100 μm. (**C**) PCR was used to verify the successful generation of the H_2_B–EGFP-labeled strains. The plasmid construct was integrated into the *A. oligospora* wild-type strain (WT: ATCC 24927). The wild-type and transformant strains are shown in WT and Lanes 1–4, respectively. Amplification with primer pairs F1/R2, F2/R3, and F4/R4 generated fragments of 2395 bp, 2607 bp, and 1002 bp, respectively. The used of primers YZ-F/YZ-R could more directly indicate the difference between the transformant and WT, that is, the transformant had a 3444 bp specific band. (**D**) Southern blotting analysis of wild-type strain (WT) and H_2_B-positive transformants. (**E**) The relative transcription levels (RTLs) of *EGFP* gene between WT and H_2_B-positive transformants at 5 days. CK is the standard (which has an RTL of 1) for statistical analysis of the RTL of gene in the H_2_B-positive transformants compared to that in the WT strain under a given condition. Error bars represent the standard deviations. Asterisks indicates a significant difference between the H_2_B-positive transformants and the WT strain (Tukey’s HSD, ** *p* < 0.01, *** *p* < 0.001). (**F**) Western blotting analysis of the positioning of EGFP and histone H_2_B was used as internal references for nuclear protein.

**Figure 2 jof-09-01183-f002:**
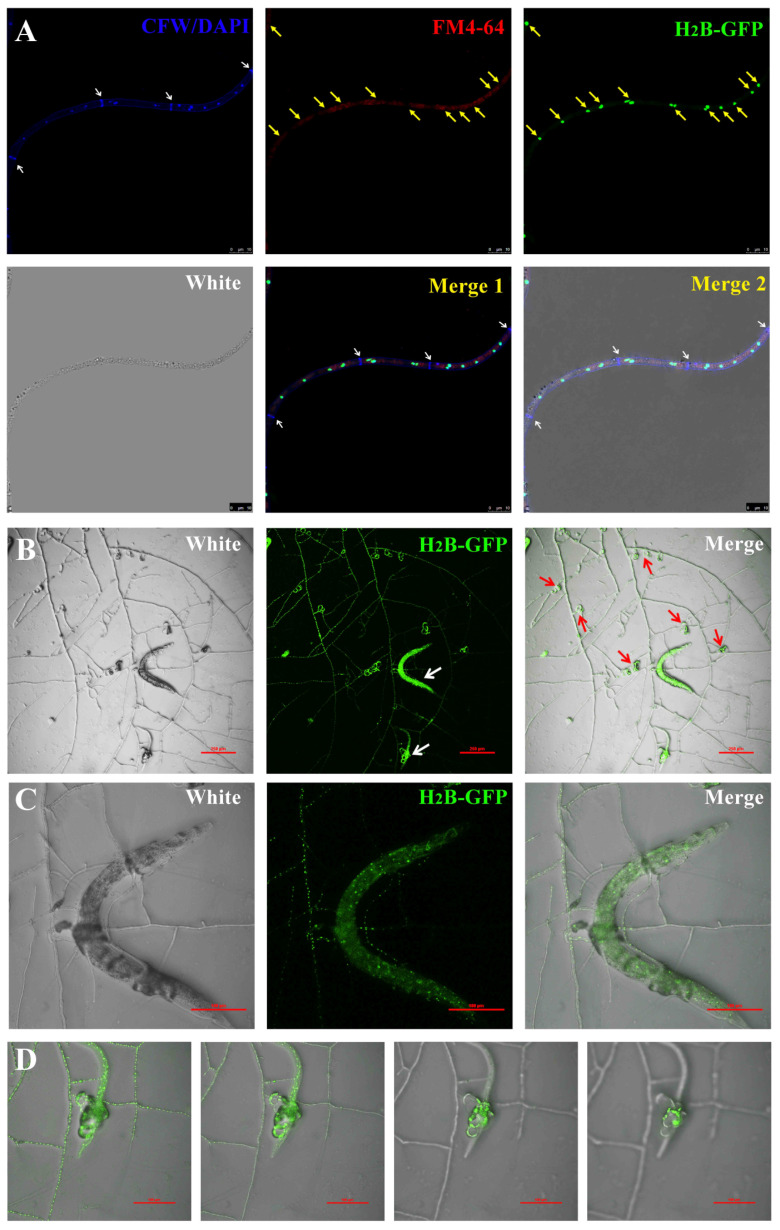
Analysis of catching nematodes with H_2_B-positive transformants. (**A**) Observation of nucleus in hyphae of H_2_B-positive transformants by staining with CFW, DAPI, and FM4-64. White arrow: septum. Yellow arrow: the position of the nucleus. Bar: 10 μm. (**B**) Observation on the capture of nematodes by H_2_B-positive transformants. White arrow: nematodes. Red arrow: traps. Bar: 250 μm. (**C**) Observation of caught nematodes. Bar: 100 μm. (**D**) Observation of traps on different planes. Bar: 100 μm.

**Figure 3 jof-09-01183-f003:**
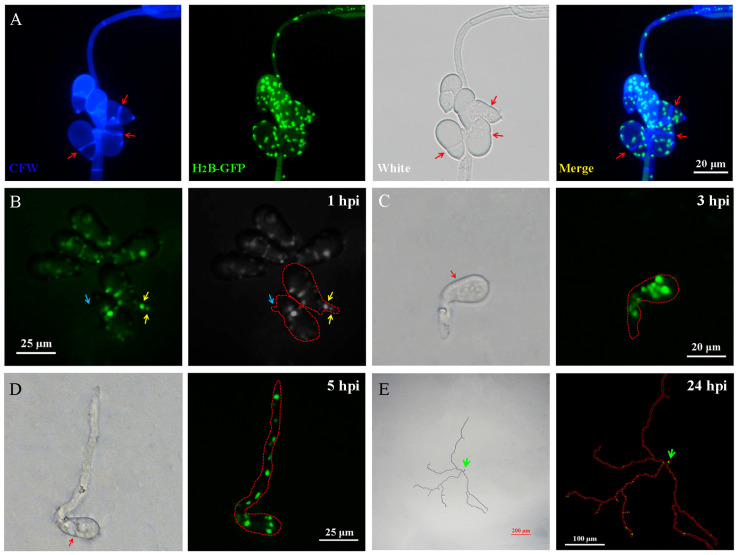
Living cell imaging of the nuclear distribution in the germination tube of *A. oligospora* spores. (**A**) Observation of nucleus in spores of H_2_B-positive transformants by staining with CFW. Bar: 20 μm. (**B**) Observation on nuclear localization of germ tubes formed by conidium at 1 h post-inoculation (hpi). Blue arrow: a nucleus was located in the germ tube. Yellow arrow: as the germ tube extended, two nuclei could be observed within it. Bar: 25 μm. (**C**) At 3 hpi, the nuclei in the elongated vegetative hyphae. Red arrow: septum. Bar: 20 μm. (**D**) At 5 hpi, more nuclei enter the hyphae and migrate towards the tip of the hyphae. Red arrow: septum. Bar: 25 μm. Red dashed line: Outline of spores and germinating hyphae. (**E**) Observation of hyphae formed by the germination of one spore during one day of incubation. Green arrow: spore. Red solid line: outline of conidia and mycelium.

**Figure 4 jof-09-01183-f004:**
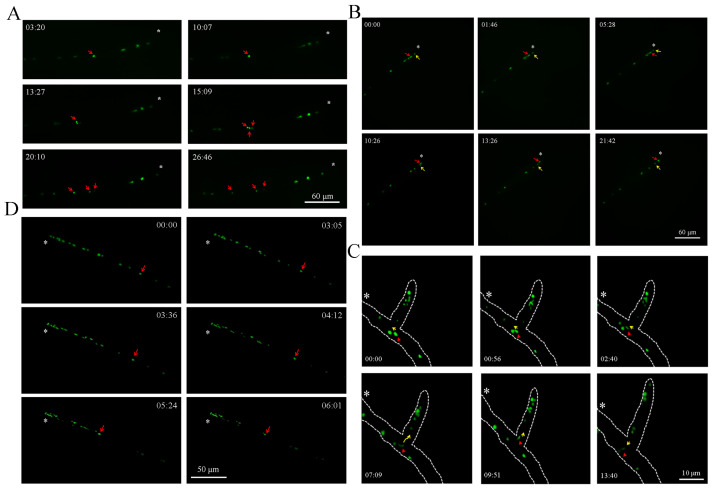
Living cell imaging of the nuclear distribution in the hyphae of *A. oligospora*. (**A**) Observation of nuclear division in mycelium. Red arrow: the nucleus associated with division. Bar: 60 μm. (**B**) Observation of cell nucleus migration at the tip of hyphae. Yellow arrow: the nucleus at the tip of the hyphae. Red arrow: the nucleus that had undergone migration. Bar: 60 μm. (**C**) Observation of nuclear migration in branching hyphae. Red triangle: migrated nucleus. Yellow arrow: the direction of nuclear migration. White dashed line: mycelium outline. Bar: 10 μm. (**D**) Observation of a large number of nuclei migrating towards the tip of the hyphae. Red arrow: a rapidly migrated nucleus. Bar: 50 μm. Asterisk: tip of mycelium. White numbers: fluorescence localization at different time points.

**Figure 5 jof-09-01183-f005:**
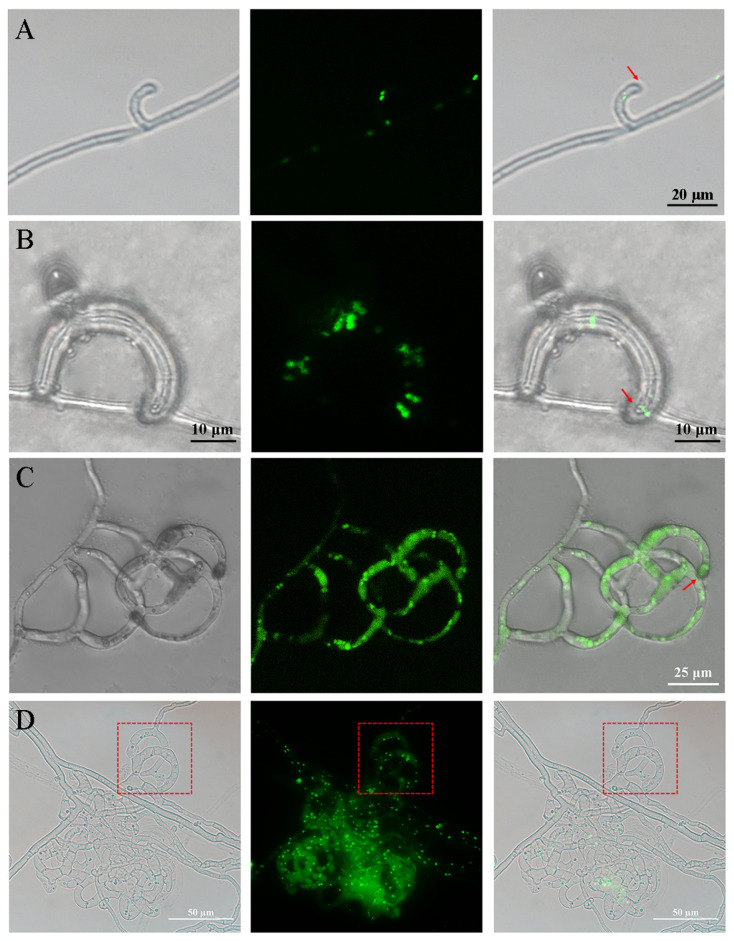
Living cell imaging of the nuclear distribution in the trap formation process of *A. oligospora*. (**A**) The localization of the nucleus at the beginning of the curvature of the hyphae. Red arrow: the tip of a curved mycelium. Red square: the connection way between the trap loops. Bar: 20 μm. (**B**) Localization of the nucleus where the mycelium bended into a semicircle. Red arrow: the tip of a curved mycelium. Bar: 10 μm. (**C**) Observation of the nucleus of immature traps that have formed multiple loops. Red arrow: the tip of a curved mycelium. Bar: 25 μm. (**D**) Observation of the nucleus of mature traps that form a large number of cell loops. Bar: 50 μm.

**Figure 6 jof-09-01183-f006:**
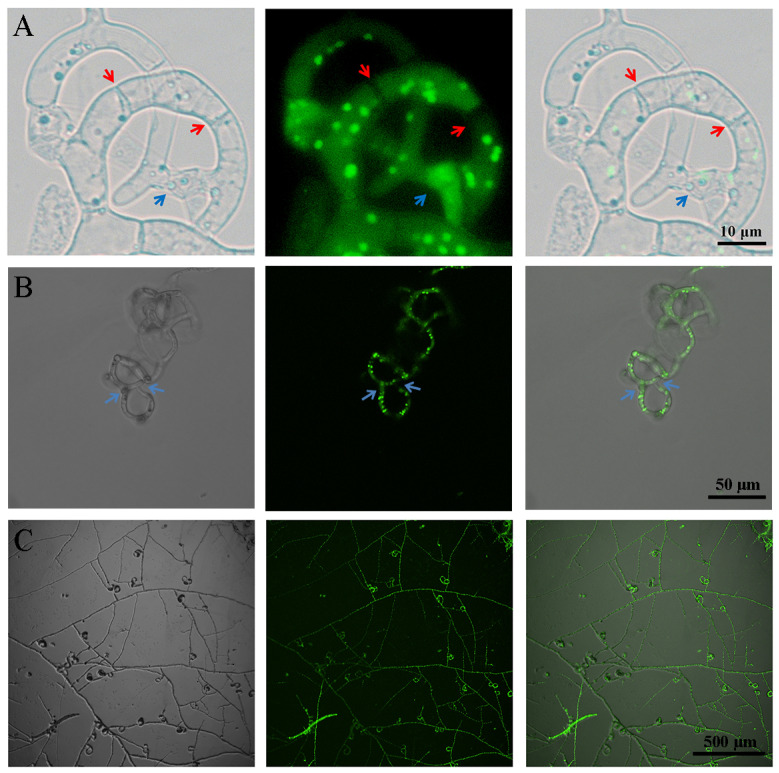
Observation of the connection mode between the fungal loops of the traps. (**A**) The connection mode between the space capture loops, which shows the image inside the red dashed box in Figure 5D. Red arrow: septum. Blue arrow: the trap loops were connected to each other through hyphae. Bar: 10 μm. (**B**) A new loop was formed on the trap loop. Blue arrow: the connection point of the newly formed loop. Bar: 50 μm. (**C**) Observation of traps in different loop forming states. Bar: 500 μm.

## Data Availability

The data for this study, both within the article and Appendix A, are readily accessible.

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
