# Peer review of "Analysis of Nuclear Dynamics in Nematode-Trapping Fungi Based on Fluorescent Protein Labeling"

_jof, 2023, doi:10.3390/jof9121183_

Round 1

Reviewer 1 Report

Comments and Suggestions for Authors

Analysis of Nuclear Dynamics in Nematode-Trapping Fungi Based on
Fluorescent Protein Labeling

The abstract and the text of the article should be written impersonally.

The key words should be different from the title.

The conclusions in the abstract should be in line 497: We speculate that the changes in the shape of nuclei and their fixation indicate the instantaneous loss of the interaction between these nuclei and cyto skeleton mechanisms during nuclear migration. In summary, these results allow us to assume that the signal that triggers nuclear migration into fungal traps is generated at the 500 beginning of the entry of the nucleus into a trap cell.

The work in the light of science has made some contribution.

three histone H2B-EGFP-la- 99 beled strains, but this was partially achieved.

The material and methods are well described as well as the results, however, more recent information on the specific application of these fungi in practice could be added to the discussion as: Braga FR, de Araújo JV. Nematophagous fungi for biological control of gastrointestinal nematodes in domestic animals. Appl Microbiol Biotechnol. 2014 Jan;98(1):71-82. doi: 10.1007/s00253-013-5366-z

Reviewer 2 Report

Comments and Suggestions for Authors

The manuscript with the title “Analysis of Nuclear Dynamics in Nematode-Trapping Fungi Based on Fluorescent Protein Labeling” investigated Athrobotrys oligospora (a nematode parasitic fungi) nuclear dynamics by histone H2B-EGFP labeling.

Abstract is succinct and clear.

Key words shall not repeat words found in the title e.g. nuclear dynamics.

Introduction is well-constructed. First paragraphs summarize the importance and diversity of useful fungi. I would suggest to a add some brief information on the taxonomic diversity NTF and specify to which phylum/order they belong. This immediately is relevant in understanding their general characteristics and placing in the tree of life.

Material and Methods are sufficiently detailed. I suggest to add a separate chapter with statistical analyses which explains what tests were performed and why, software used and anything else relevant.

Results and Discussion are eloquent and well-presented. Images have good quality.

Best regards.

Comments on the Quality of English Language

fine English style improvement could help.

Reviewer 3 Report

Comments and Suggestions for Authors

The manuscript titled "Analysis of Nuclear Dynamics in Nematode-Trapping Fungi Based on Fluorescent Protein Labeling" is appropriate for the journal. The research topic provides an original and relevant contribution to living cellular biology of processes such as spore germination, hyphal extension and fusion, essential in the physiology of filamentous fungi. The study of Arthrobotrys oligospora, a nematophagous fungus, from this perspective helps to better understand its mechanism as a biological control agent. The authors carried out extensive research work. At the end of the document you need a general conclusion of your research.

The article may be accepted for publication in its current format.
